# Effect of Low Copper Addition to As-Forged 304 Stainless Steel for Dental Applications

Ker-Kong Chen [1,2,†], Chih-Yeh Chao [3,†], Jeng-Huey Chen [1,2], Ju-Hui Wu [1,4], Yen-Hao Chang [5] and Je-Kang Du [1,2,*]

1   Department of Dentistry, Kaohsiung Medical University Hospital, Kaohsiung 80708, Taiwan; enamel@kmu.edu.tw (K.-K.C.); 720101@kmu.edu.tw (J.-H.C.); wujuhui1020@gmail.com (J.-H.W.)
2   School of Dentistry, College of Dental Medicine, Kaohsiung Medical University, Kaohsiung 80708, Taiwan
3   Department of Mechanical Engineering, National Pintung University of Science and Technology, Pingtung 91201, Taiwan; cychai@mail.npust.edu.tw
4   Department of Oral Hygiene, College of Dental Medicine, Kaohsiung Medical University, Kaohsiung 80708, Taiwan
5   PhD Program, School of Dentistry, College of Dental Medicine, Kaohsiung Medical University, Kaohsiung 80708, Taiwan; edward590198@gmail.com
*   Correspondence: dujekang@gmail.com; Tel.: +886-7-3121101 (ext. 7003)
†   These authors contribute equally to this work.

**Abstract:** The aim of this study was to investigate the effect of incorporating low copper (0, 0.5, 1, 1.5, and 2 wt.%) additions into as-forged AISI 304 stainless steel (304SS). The microstructures and mechanical properties of the steel were examined using scanning electron microscopy and a universal testing machine. The antibacterial properties of the Cu-bearing 304SS specimens were investigated using *Escherichia coli*. Each specimen was soaked in artificial saliva to detect the release of copper ions through inductively coupled plasma atomic emission spectrometry. The addition of copper had no significant effect on the microstructure of the as-forged Cu-bearing 304SS, but it slightly increased its maximum tensile strength. The antibacterial rate of the as-cast and as-forged 304SS with 2 wt.% Cu was over 80%, which corresponded to an increase in the release of copper ions. This study demonstrates that low-Cu-content stainless steel can reduce bacteria and can be a suitable material for the oral environment because of the low release of Cu ions.

**Keywords:** Cu-bearing stainless steel; microstructures; mechanical properties; antibacterial properties



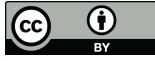

## 1. Introduction

Orthodontic appliances, orthodontic wires, and miniscrew implants (MSIs) are common and are widely used in orthodontic clinics [1,2]. In recent years, MSIs have been frequently used as bone anchors for orthodontic treatment. In addition to improving the treatment mode of orthodontics, MSIs can also greatly shorten the treatment time. MSIs are generally made of titanium alloys [3]. Titanium is biocompatible and allows for a high bone–implant contact ratio (osseointegration) between dental implants and the host bone after healing [4]. From the point of view of the clinician, an ideal MSI requires immediate loading after implantation, sufficient mechanical rigidity, and resistance to corrosion. After the orthodontic treatment is completed, the MSI must be easy for the clinician to remove without breaking. However, the high degree of osseointegration required for dental implants is not a requirement for orthodontic MSIs to function as anchorage devices [5]. Unlike titanium alloys, stainless steel (SS) screws tend to develop primary stability (a mechanical locking interface) between the screw and bone, which lowers the removal torque, leading to easier retrieval [6].

There are four major differences between MSIs and general dental implants. One is that MSIs placed in the alveolar bone must be partially exposed outside the oral mucosa, which increases the risk of infection. Second, MSIs need to be immediately loaded and

placed in the alveolar bone. Third, the direction of the force is usually lateral, and the implant needs to be removed after the orthodontic treatment is completed. Fourth, MSIs are usually small in size and diameter.

The use of MSIs requires that the existence of multiple microorganisms in the oral cavity be considered. Colonies of these microorganisms form a biofilm in the oral cavity, which can easily cause bacteria to accumulate and attach to the surfaces of instruments and prostheses, causing inflammation and infection of the soft tissues [7]. Some scholars have pointed out that tissue inflammation, infection, and peri-implant inflammation may increase the failure rate of MSI implantation by 30% in its initial stage [8]. To solve these problems, stainless steel can acquire antibacterial functions through methods including surface treatment [9], amorphous treatment [10], and alloy design [11]. However, as surface spraying can only result in short-term antibacterial effects, alloy design is a promising method for obtaining long-term antibacterial properties [11].

Copper exists in trace amounts in the human body (100–200 mg) [12]. The upper limit of daily copper intake in the human body is 0.7–2 mg/d, depending on the age group [12]. A moderate amount of copper can promote the production of collagen, bones, blood vessels, and skin in the human body [11]. However, an excessive intake of copper causes cell damage or death, which harms the human body and causes diseases such as Wilson's disease and Alzheimer's disease [13]. Although copper-containing stainless steel has good mechanical and antibacterial properties, it has development potential in the field of biomedical materials that is yet to be exploited. However, the release of copper ions should still be controlled so as to ensure the safety of copper-containing stainless steel for human use. Furthermore, copper has been proven to be an effective antibacterial element [14–17]. Therefore, many studies have been conducted on copper-containing stainless steel. However, because copper is a stable element of austenitic iron in stainless steel, it has a grain refinement effect. Therefore, the addition of copper can strengthen the mechanical properties of stainless steel [18]. In terms of improving the mechanical properties of stainless steel, forging [18] and heat treatment [19] are also used to meet the strength requirements in stainless steel for use in different types of medical equipment.

Austenitic stainless steel has numerous excellent properties, such as nonmagnetism, high strength, good corrosion resistance [20], and superior workability [12]. It is widely used in industrial applications and is suitable for biomedical applications. The AISI 304 alloy is one of the most popular stainless steels. Zhang et al. [11] added 3.9 wt% copper to 304 stainless steel. The experimental steel was solution treated and then aged so as to obtain Cu-rich phase precipitation in the matrix, which afforded the steel a strong antibacterial function against *Porphyromonas gingivalis* (*P. gingivalis*). However, the corrosion resistance of copper-containing stainless steel is more controversial. Some studies have reported that copper-added stainless steel is prone to local corrosion phenomena [13], such as pitting corrosion [21], stress corrosion [22], and grain boundary corrosion [23]. Tong et al. [24] added 2.5–3.5 wt% copper to 316 stainless steel. Their experimental results showed that 316 copper-containing stainless steel could improve mechanical strength after solid solution and aging heat treatment. However, the corrosion resistance of 316 stainless steel that contains copper is worse than that of pure 316 stainless steel. Although the precipitation of the Cu-rich phase can improve the antibacterial properties, Hong et al. [14] indicated that the presence of copper precipitates affects the stability of the passive film on the surface of stainless steel and reduces its corrosion resistance. Therefore, the copper content of current, common copper-containing austenitic stainless steel is approximately 3 to 5 wt% [16,24–26].

Even though the biocompatibility of 5 wt% copper with stainless steel is within the allowable range, there are still concerns about excessive dissolution in the oral cavity in the long term. The intake of excessive copper leads to severe liver damage (cirrhosis), owing to copper-induced oxidative damage in the liver and other tissues [27]. Therefore, in this study, we mainly investigated the antibacterial and mechanical properties of forged 304

stainless steel by adding a low copper content (0.18, 0.54, 1.05, 1.80, and 2.05 wt%). We also measured the release of copper ions by soaking the materials in artificial saliva.

## 2. Materials and Methods

### 2.1. Specimen Preparation

In this study, 0.18, 0.54, 1.05, 1.80, and 2.05 wt% copper was added to austenitic 304 stainless steel (Table 1), which was provided by the Datian Precision Industry Co., Ltd. (O-TA Company, Shenzhen, China). The alloy was melted in a vacuum melting furnace (Ultimate Materials Technology Co., Ltd., Hsinchu, Taiwan)at 1700 °C. The smelted stainless steel block was homogenized at 1500 °C for 1 h. In addition, forging at 1300 °C was performed to produce plates with a thickness of $3 \pm 0.1$ mm, which was followed by solution treatment and water cooling at 880 °C for 1 h. The resulting castings and forgings were all made into standard square tensile test bars of $10 \times 10 \times 2$ mm$^3$ so as to facilitate the subsequent experiments.

**Table 1.** Chemical composition (wt%) of the experimental Cu-bearing 304 stainless steel.

| Specimens | Fe | Cr | Ni | Si | C | Mn | S | P | Cu |
|---|---|---|---|---|---|---|---|---|---|
| 304 | Bal. | 18.14 | 8.11 | 0.46 | 0.02 | 0.80 | 0.03 | 0.04 | - |
| 304–0.2Cu | Bal. | 18.21 | 8.13 | 0.43 | 0.02 | 0.88 | 0.03 | 0.03 | 0.18 |
| 304–0.5Cu | Bal. | 18.29 | 8.02 | 0.53 | 0.03 | 0.93 | 0.03 | 0.04 | 0.54 |
| 304–1.0Cu | Bal. | 18.16 | 8.09 | 0.55 | 0.03 | 0.95 | 0.02 | 0.03 | 1.05 |
| 304–1.8Cu | Bal. | 18.10 | 8.03 | 0.50 | 0.02 | 0.91 | 0.03 | 0.03 | 1.82 |
| 304–2.0Cu | Bal. | 18.13 | 8.03 | 0.51 | 0.03 | 0.83 | 0.03 | 0.04 | 2.05 |

### 2.2. Microstructure

The as-forged specimens were prepared by grinding using a series of 150#- to 1500#-grit SiC sandpapers. Then, alumina powder ($Al_2O_3$) and silicon dioxide ($SiO_2$) were used for polishing. The specimens were corroded in a mixed solution containing 30% nitric acid ($HNO_3$) + 10% hydrochloric acid (HCl) + 5% hydrofluoric acid (HF) + 55% alcohol for 3 min. Water and alcohol were used to ultrasonically clean the corroded specimens for 10 min. A scanning electron microscope (SEM; JEOL JSM-6380, Tokyo, Japan) was used to observe the microstructure under 15 kV.

### 2.3. Mechanical Properties

The as-forged tensile test specimen in this study was made according to the standard size specifications of the American Society for Testing and Materials. Its neck diameter was 6.25 mm, length was 105 mm, and diameter was 12 mm. Then, a universal testing machine (E45.305, Eden Prairie, MN, USA) was used to perform the tensile test at a rate of 3 mm/min. The hardness was measured using a Vickers Hardness machine (MVK-H11, Akashi, Tokyo, Japan) to randomly measure five specimens under a load of 500 g. Each specimen was measured at three points, and the average was obtained.

### 2.4. Antibacterial Testing

The antibacterial properties of as-cast and as-forged 304 SS and Cu-bearing 304 SS were evaluated according to the JIS Z2801–2000 specifications [28]. The experimental specimens were sterilized using laboratory autoclaves (MLS-3781L, Tokyo, Japan). Then, 0.4 mL of an *Escherichia coli* (*E. coli*, ATCC-25922, Food Industry Research and Development Institute, Hsinchu, Taiwan) suspension with a density of $10^6$ CFU/mL was dripped onto the specimens (with a surface area of $10 \times 10$ mm$^2$), as well as onto a control group (304 SS). The specimens were washed with phosphate-buffered saline after standing in the suspension for 24 h. The bacterial solution was diluted to $10^6$ and $10^7$, and 100 μL was applied to each agar plate. Afterwards, the plates were placed in orbital shaking incubators. The number of remaining colonies on the medium was calculated after 24 h.

The experiment was repeated three times, and the formula for calculating the antibacterial rate was as follows:

$$\text{Antibacterial rate (AR)} = (N_{304} - N_{304-Cu})/N_{304} \times 100\%$$

where $N_{304}$ and $N_{304\text{-}Cu}$ are the average numbers of bacterial colonies on the 304SS and 304-Cu specimens, respectively.

### 2.5. Copper Ion Release

The as-cast and as-forged 304SS and Cu-bearing 304 were immersed in artificial saliva [29] with a pH value of 7.5 at 37 °C, per ISO Standard 10993–12:2002, for 1, 2, and 3 days. The concentration of Cu ions released was analyzed using inductively coupled plasma atomic emission spectrometry (PerkinElmer Inc.—Optima 2100 DV, Waltham, MA, USA) with an accuracy of 0.01 mg/L.

## 3. Results

### 3.1. Microstructure Observation

Typical SEM images of the alloys subjected to solution treatment are shown in Figure 1. The microstructure of the as-forged Cu-bearing 304SS was a single austenite ($\gamma$) phase. With the addition of Cu, it was found that all specimens presented similar microstructures, implying that the Cu addition had no obvious influence on the microstructures. Moreover, the grain sizes of Cu-bearing 304SS with different Cu additions were also similar, and the average grain sizes were approximately 25 μm, indicating that a low Cu addition played no apparent role in determining the grain size.

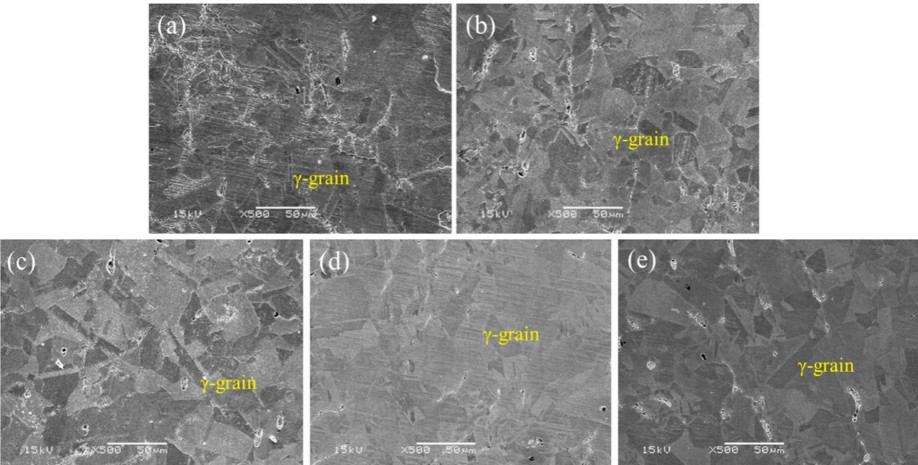

**Figure 1.** SEM images of forged 304 stainless steel with different copper contents: (**a**) 304–0.2Cu, (**b**) 304–0.5Cu, (**c**) 304–1Cu, (**d**) 304–1.8Cu, and (**e**) 304–2Cu.

### 3.2. Mechanical Properties

The detailed mechanical properties, including the ultimate tensile strength (UTS), yield strength (YS), elongation, and hardness under different Cu additions, are shown in Table 2. It is noted that the increase in Cu content had a slight influence on the UTS and YS of Cu-bearing 304SS. The increment in UTS and YS was approximately 30 MPa relative to those of 304SS. However, the addition of Cu had no effect on elongation. The hardness of the Cu-bearing 304SS with different Cu contents showed no obvious variation.

**Table 2.** Mechanical properties of the forged 304 stainless steel with different copper contents.

| Specimen | Ultimate Tensile Strength (UTS, MPa) | Yield Strength (YS, MPa) | Elongation (%) | Hardness (Hv) |
|---|---|---|---|---|
| 304 | 591.6 | 355.2 | 57.1 | 213.4 |
| 304–0.2Cu | 590.2 | 350.3 | 58.4 | 212.6 |
| 304–0.5Cu | 601.7 | 368.3 | 58.1 | 213.2 |
| 304–1.0Cu | 627.4 | 395.2 | 55.3 | 216.4 |
| 304–1.8Cu | 632.8 | 400.7 | 54.9 | 217.5 |
| 304–2.0Cu | 639.5 | 411.0 | 55.1 | 220.9 |

### 3.3. Antibacterial Testing

Figure 2 shows the antibacterial properties of the alloy in this study and *E. coli* co-cultured for 1 d. The figure shows that the antibacterial rate of cast 304 and forged 304 stainless steel increased with the copper content. This shows that higher copper content results in better antibacterial properties, and the antibacterial properties of the two show the same trend. Among them, the antibacterial rates of cast 304 and forged 304 stainless steel reached 82.6% and 80.9% for 2 wt.% Cu copper content.

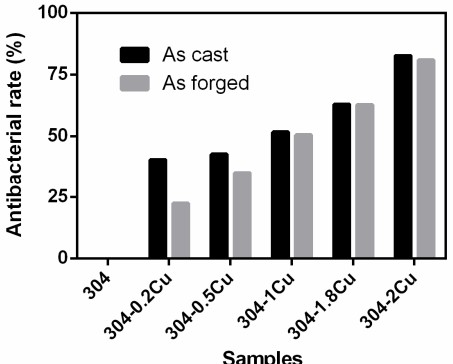

**Figure 2.** Colonization of *E. coli* after 24 h coculture with a change in Cu content.

### 3.4. Copper Ion Release

The copper ion release results are shown in Figure 3. Regardless of the forging or casting of copper-containing 304 stainless steel, the accumulated copper ion concentration increased with the copper content in the physiological environment. The data also revealed that the concentration of released Cu ions increased with the immersion time. The release concentrations of Cu ions from cast and forged 304-Cu specimens were 0.13–0.24 and 0.10–0.19 mg/L over three days, respectively. Furthermore, it is noted that as-cast 304Cu exhibited a relatively higher release concentration than the as-forged.

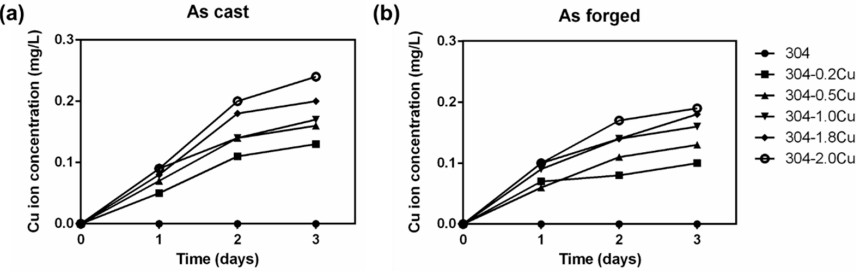

**Figure 3.** Release concentrations of copper ions from 304Cu stainless steel over 1, 2, and 3 days: (**a**) as cast and (**b**) as forged.

## 4. Discussion

Stainless steel is widely used in medical equipment. In the field of dentistry, there are many challenges because of the complex oral environment, especially bacteria in the oral cavity. In addition, medical devices exposed to the oral environment, such as orthodontic wires, need to withstand tension in order to maintain the orthodontics for ideal positioning of the teeth. Therefore, the combination of antibacterial properties, biocompatibility, and mechanical properties is very important. Copper is used as an alloying element added to steel. Its main purpose is to improve the mechanical properties, corrosion resistance, and cold deformation of steel [13,30]. Furthermore, $Cu^{2+}$ is similar to $Ag^+$, which has a high affinity for the thiol groups contained in some enzymes and amino acids. It can disrupt the bacterial metabolism or generate reactive oxygen species through redox reactions by binding to these thiols [31]. Therefore, it can endow stainless steel with an antibacterial ability [32]. In the present study, a low-Cu-bearing 304SS was developed. The microstructures, antibacterial properties, mechanical properties, and Cu ion release concentrations of as-cast and as-forged Cu-bearing 304SS were investigated.

A higher Cu content in the passive film lowers the stability of the oxide film because of the reaction of $Cu + 2e \rightarrow Cu^{2+}$ and enhances the flux of the charge transfer [24]. Thus, the increase in Cu content in the passive film significantly decreased the corrosion resistance of the Cu-bearing SS. Additionally, copper-containing stainless steel precipitates a Cu-rich phase owing to the supersaturated copper atoms. Ren [12] indicated that 304 stainless steel containing copper precipitates the Cu-rich phase after long-term aging treatment. In addition, different heat treatments can also control the amount of precipitates formed and play an important role in the resulting antibacterial properties [33]. However, owing to the formation of a potential difference between the Cu-rich phase and the matrix, the galvanic corrosion rate is accelerated, and the corrosion resistance of stainless steel is reduced [12,24]. Therefore, in this study, we only used low-copper and solid-solution treatments to maintain corrosion resistance. The addition of approximately 0.2–2.0 wt.% copper has no effect on the microstructure after forging. It was observed from the SEM image that there was no precipitation at the base or boundary. Copper is a stable element in the austenitic phase of stainless steel; it can replace nickel and also has a solid solution strengthening effect. Therefore, both UTS and YS increased with the copper content and showed similar trends.

The antibacterial performance of copper is weaker than that of silver; therefore, a relatively high copper content is needed to obtain the required antibacterial performance. However, an upper limit for the safe daily intake of copper exists [34] in the human body; therefore, the addition of copper needs to be carefully considered. In the experimental results for copper ion release in this study, it was found that the release amount of cast 304 stainless steel with low copper (0.2–2.0 wt.%) was higher than that of the forged steel. A possible reason for this is that after forging, stainless steel can obtain better corrosion resistance than after casting; therefore, the copper ions are less likely to be released into the electrolyte. In particular, the highest release of copper ions was only 0.24 mg/L for the alloy used in this study. This is below the upper limit of daily copper intake in the human body. This shows that the alloy has no direct safety concerns. In addition to the antibacterial properties of copper, a low concentration (13.3 μM) of copper is required for proteases in organisms. An appropriate amount of copper can maintain the stable growth of collagen and skin in the human body, and reduce the occurrence of cardiovascular disease [35]. Combining the results of the antibacterial property and Cu ion release tests, it can be deduced that the addition of metallic Cu into the 304-Cu SS could provide bivalent Cu ions, which are key antibacterial agents. Xi et al. [24] explained that when 316L–2.5Cu was solution treated at 1100 °C for 30 min and 700 °C for 6 h, a higher number and density of spherical Cu-rich precipitates, with a diameter of approximately 10 nm, were homogeneously distributed within the matrix compared with in this study. The alloy had an *E. coli* antibacterial rate of 94.5%. In addition, Zhang et al. [14] added 3.9 wt% copper to 304 stainless steel followed by a 1040 °C solution treatment for 30 min and a 700 °C aging treatment for 6 h so as to precipitate the copper-rich phase. *P. gingivalis* was used for the

antibacterial experiments in their study. The antibacterial properties of copper-containing stainless steel reached 100% after 10–12 h. This shows that the copper-rich phase plays an important role, but that it is necessary to further evaluate the antibacterial and corrosion resistance of the copper-rich phase.

Numerous studies have shown that regardless of whether alloys are stainless steel or titanium, the appropriate addition of copper can bring antibacterial properties to the material [28,32,36,37]. However, considering the actual application conditions, the oral environment is complex, and the material is soaked in saliva for a long time. Thus, antibacterial properties that are too great (large amounts of Cu ions released) may not be suitable for oral biomedical materials. In addition, a large copper addition and long-term heat treatment will increase the cost of production. Therefore, it may be possible to modify the alloy design at the application level in order to meet the needs of clinical use in the development of medical equipment. This study confirms that cast or forged stainless steel with a 2 wt.% Cu content can reach an antibacterial rate of more than 80%, indicating that a low copper content can still have distinct advantages and can maintain a balanced relationship between antibacterial properties and corrosion resistance.

## 5. Conclusions

The addition of Cu in the present study had no influence on the microstructure of the as-forged 304SS. The YS and UTS of the forged 304-Cu SS was increased under different Cu contents, with corresponding solution strengthening. The antibacterial rate of the as-forged 304SS with 2 wt.% Cu was over 80%, which corresponded to an increased release of copper ions. The amount of copper ions released showed no toxicity to humans in the oral environment. The current study indicates that antibacterial Cu-bearing 304 SS is a potential material for the oral environment.

**Author Contributions:** Conceptualization, K.-K.C., J.-H.C., and C.-Y.C.; methodology, J.-K.D. and C.-Y.C.; software, J.-H.W. and Y.-H.C.; validation, J.-K.D. and J.-H.C.; formal analysis, C.-Y.C. and K.-K.C.; investigation, J.-K.D. and Y.-H.C.; resources, J.-H.C. and J.-H.W.; data curation, J.-H.W.; writing—original draft preparation, J.-K.D., Y.-H.C., and C.-Y.C.; writing—review and editing, J.-K.D., C.-Y.C., and Y.-H.C.; visualization, J.-H.C.; supervision, K.-K.C.; project administration, K.-K.C.; funding acquisition, K.-K.C. and J.-K.D. All authors have read and agreed to the published version of the manuscript.

**Funding:** This study was financially supported by funding from the Ministry of Science and Technology (MOST 106–2314-B-037–013 and MOST 107–2314-B-037–46), and the Kaohsiung Medical University Hospital (KMUH104-4M47, KMUH 106–6R73 and KMUH 108–8M62).

**Conflicts of Interest:** The authors declare no conflict of interest.

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
