# Peer review of "Effect of Low Copper Addition to As-Forged 304 Stainless Steel for Dental Applications"

_metals, doi:10.3390/met11010043_

Round 1

Reviewer 1 Report

The paper focuses on an important area of research. Better antibacterial properties of implants are highly needed and the paper give some important insights in the field. 

The paper could need some minor english editing but is well presented. 

Author Response

Comments and Suggestions for Authors:

Reviewer I

The paper focuses on an important area of research. Better antibacterial properties of implants are highly needed and the paper give some important insights in the field.

The paper could need some minor english editing but is well presented.

Answer: Thank you for your comments and suggestions. The manuscript has been edited by a professional English editing group to check grammar error. Besides, we have attached the CERTIFICATE OF ENGLISH EDITING.

Thank you for your consideration. We hope our manuscript is suitable for publication in your journal.

Sincerely,

Je-Kang Du

Reviewer 2 Report

You have noted the ressentiments against copper at several places. I consider this one of the most severe objections against application of copper (in fact: active Cu2+) and therefore recommend to stress this more explicitely in your conclusion that solving Cu2+ ions out of a depot of metallic copper (here: alloyed in stainless steel) can be used for long-term therapy. 

Your measurements to determine the concentration of dissolved Ch2+ ions neglect one very important issue, namely the time-dependent solution behavior:

1) Is there an induction period?

2) Does the dissolution saturate (i. e. do you observe a constant dissolution  rate) or will it decrease eventually, for example due to passivation in alkaline /oxidative environment) I would expect to see a passivation in alkaline environmet. 

In your graphs (Figs. 2+3), you use the font Helvetica/Aria. That's fine. But you should use no more than 5 digits at every axis. It would be perfect if you would add axes at the top and to the right. And same style. In  Fig. 2, you use bold characters, in Fig. 3, you use normal characters. Have you tried to determine the chemical equilibrium of Cu --> Cu2+(aq.)  at 37 °C in alkaline environment (pH 7.5 or so)? Your paper would gain a higher (scientific) level. Up to now, it is still describing your experiments. That's fine, but it could be better. 

I would strongly recommend to make these experiments. 

Author Response

Comments and Suggestions for Authors:

Reviewer II

You have noted the ressentiments against copper at several places. I consider this one of the most severe objections against application of copper (in fact: active Cu2+) and therefore recommend to stress this more explicitely in your conclusion that solving Cu2+ ions out of a depot of metallic copper (here: alloyed in stainless steel) can be used for long-term therapy.

Your measurements to determine the concentration of dissolved Ch2+ ions neglect one very important issue, namely the time-dependent solution behavior:

  • Is there an induction period?

Answer: Thank you for your deep concern. According to the JIS Z2801-2000 specifications, the process of antibacterial test is to culture bacteria on the specimens for "one day", and then cultivate the survival bacteria remaining on the surface of specimens to calculate the antibacterial rate. Based on the results of antibacterial experiments, it was confirmed that the specimens had antibacterial properties after one day. This means that the induction period will be shorter than one day. Clinically, due to the complexity of the human body, the interaction between bacteria and the human body is affected by many factors, such as the immune system. And “one day” will be shorter than the time for bacteria to react with the human body (usually 48~72h).

  • Does the dissolution saturate (i. e. do you observe a constant dissolution rate) or will it decrease eventually, for example due to passivation in alkaline /oxidative environment) I would expect to see a passivation in alkaline environment.

Answer: We appreciate this helpful comment from the reviewer. The immersion test was conducted in artificial saliva (adjusted pH=7.5) at 37°C. In the previous version, the Cu ions was measured at 3rd day. In the present version, we supplemented the released concentration of Cu ions in artificial saliva during 1, 2 and 3 days. It can be found that the accumulated copper ions are relatively increasing. However, the daily release of copper ions decreases with the immersion time, which is consistent with the trend in other research (as reviewer said passivation). This information has been supplemented at line 141-144 Page 4 of revised manuscript and marked with red.

In your graphs (Figs. 2+3), you use the font Helvetica/Aria. That's fine. But you should use no more than 5 digits at every axis. It would be perfect if you would add axes at the top and to the right. And same style. In Fig. 2, you use bold characters, in Fig. 3, you use normal characters.

Answer: Thank you for your pointing out. Figure 2 and 3 have been remade. The font and word style have been revised to be consistent.

Have you tried to determine the chemical equilibrium of Cu --> Cu2+(aq.) at 37 °C in alkaline environment (pH 7.5 or so)? Your paper would gain a higher (scientific) level. Up to now, it is still describing your experiments. That's fine, but it could be better.

Answer: Thank you for your kind suggestions. The used artificial saliva was adjusted to 7.5 to conduct the Cu ion release test. This has been supplemented in the section 2.5. And, the released Cu2+ from the surface of each sample was measured at 1, 2 and 3 days. This information has been supplemented at line 174-179 Page 5 of revised manuscript and marked with red.

I would strongly recommend to make these experiments.

Answer: We appreciate this helpful comment from the reviewer. We have added the release concentration of copper ions in each group of 304Cu at different time points. This information has been supplemented at line 141-144 Page 4 and line 174-179 Page 5 of revised manuscript and marked with red.

Thank you for your consideration. We hope our manuscript is suitable for publication in your journal.

Sincerely,

Je-Kang Du

Reviewer 3 Report

This is indeed an interesting and valuable paper dealing with antibacterial properties of SS. The authors present that increase in Cu in the SS significantly influences on antibacterial properties, which is of high importance for medical implants as well as surgical tools. The author also studied the influence of Cu ion release from SS in artificial saliva. Results show, as expected, that higher Cu content is connected also with higher ion release. The presented results show high potential, as antibacterial activity against E. Coli was observed, especially for the forged steels. In my opinion the topic is highly relevant and I would recommend this article to be published after minor revision.

Minor comments:

  • The forged and cast SS are not well discussed and presented- in the methods part there is no mentioning on which samples the analysis was conducted. In the table for tensile tests the information on which SS samples analysis was done is not given- probably it would be valuable to add both the cast and forged- or the author should comment – if there are any differences. Similar goes for the microstructure. This seems to be important as the results of ion release and antibacterial properties show significant difference.
  • It would be valuable- for future studies also to study the surface chemistry of these materials, maybe also depth profile. Some comments on this could be added- with regards to current state of the art.

Author Response

Comments and Suggestions for Authors:

Reviewer III

This is indeed an interesting and valuable paper dealing with antibacterial properties of SS. The authors present that increase in Cu in the SS significantly influences on antibacterial properties, which is of high importance for medical implants as well as surgical tools. The author also studied the influence of Cu ion release from SS in artificial saliva. Results show, as expected, that higher Cu content is connected also with higher ion release. The presented results show high potential, as antibacterial activity against E. Coli was observed, especially for the forged steels. In my opinion the topic is highly relevant and I would recommend this article to be published after minor revision.

Minor comments:

The forged and cast SS are not well discussed and presented- in the methods part there is no mentioning on which samples the analysis was conducted. In the table for tensile tests the information on which SS samples analysis was done is not given- probably it would be valuable to add both the cast and forged- or the author should comment – if there are any differences. Similar goes for the microstructure. This seems to be important as the results of ion release and antibacterial properties show significant difference.

Answer: Thank you for your deep concern. Since most of the stainless steels use forging as the main processing method, this study is mainly to investigate the influence of different Cu content on the microstructure and mechanical properties after forging.

The reason why the casting and forging materials are selected for the antibacterial experiment and release test at the same time is to confirm whether the material is only affected by the change of copper in the case of casting. It can be determined that the material has no serious segregation.

  • It would be valuable- for future studies also to study the surface chemistry of these materials, maybe also depth profile. Some comments on this could be added- with regards to current state of the art.

Answer: Thank you for your kind suggestions. For follow-up experiments, we will conduct your suggested experiments for further analysis.

Thank you for your consideration. We hope our manuscript is suitable for publication in your journal.

Sincerely,

Je-Kang Du
